# FlashVSR: Towards Real-time Diffusion-Based Streaming Video Super-Resolution

## Abstract

Diffusion models have recently advanced video restoration, but applying them to real-world video super-resolution (VSR) remains challenging due to high latency, prohibitive computation, and poor generalization to ultra-high resolutions. Our goal in this work is to make diffusion-based VSR practical by achieving efficiency, scalability, and real-time performance. To this end, we propose **FlashVSR**, the first diffusion-based one-step streaming framework towards real-time VSR. FlashVSR runs at ~17 FPS for $768 \times 1408$ videos on a single A100 GPU by combining three complementary innovations: (i) a train-friendly three-stage distillation pipeline that enables streaming super-resolution, (ii) locality-constrained sparse attention that cuts redundant computation while bridging the train–test resolution gap, and (iii) a tiny conditional decoder that accelerates reconstruction without sacrificing quality. To support large-scale training, we also construct **VSR-120K**, a new dataset with 120k videos and 180k images. Extensive experiments show that FlashVSR scales reliably to ultra-high resolutions and achieves state-of-the-art performance with up to $\sim 12\times$ speedup over prior one-step diffusion VSR models. We will release code, models, and the dataset to foster future research in efficient diffusion-based VSR.

## 1 Introduction

Video super-resolution(VSR) has wide application in smartphone photography, social networks, and live streaming. It recovers a high-quality video from an original low-quality video, either because of imperfect capturing or compression losses. Many VSR methods have been proposed recently to reconstruct high-resolution (HR) videos from such degraded LR inputs (Wang et al., 2019; Chan et al., 2021; 2022a; Cao et al., 2021; Liu et al., 2022a; Zhou et al., 2024; Yang et al., 2024a; Xie et al., 2025; Li et al., 2025). Recently, with the rapid development of video generation models, researchers have found that a pretrained diffusion model can also significantly improve the visual quality of video restoration (Wang et al., 2025b;a; Xie et al., 2025; Chen et al., 2025b). Still, as mobile video and online streaming have become popular, there is a much higher demand for a VSR system that can handle high-resolution and infinitely long videos in real time.

However, achieving high-resolution, high-quality, and real-time streamable video super-resolution remains highly challenging, particularly for diffusion-based VSR. We identify three major obstacles: (1) high lookahead latency of chunk-wise processing. Due to memory constraints, most approaches divide long videos into overlapping chunks and process them independently, introducing redundant computation on overlapped frames and a high *lookahead latency* before the entire chunk finishes processing; (2) high computational cost of dense 3D attention. For better visual quality, most video generation models use full spatiotemporal attention, which has quadratic complexity in resolution, making it prohibitively expensive for long, high-resolution videos; (3) a potential train–test resolution gap. Most attention-based VSR models are trained on medium-resolution videos but often degrade when applied to higher resolutions (*e.g.*, 1440p). Our analysis shows this gap stems from mismatched positional encoding ranges between training and inference.

In this work, we handle all three of these challenges and build FlashVSR, the first diffusion-based video super-resolution model towards real-time and streamable processing. As shown in Fig. 1, FlashVSR achieves near real-time inference at 17 FPS at a resolution of $768 \times 1408$ on a single A100 GPU. This is significantly faster than all other diffusion-based VSR methods, including STAR (Xie

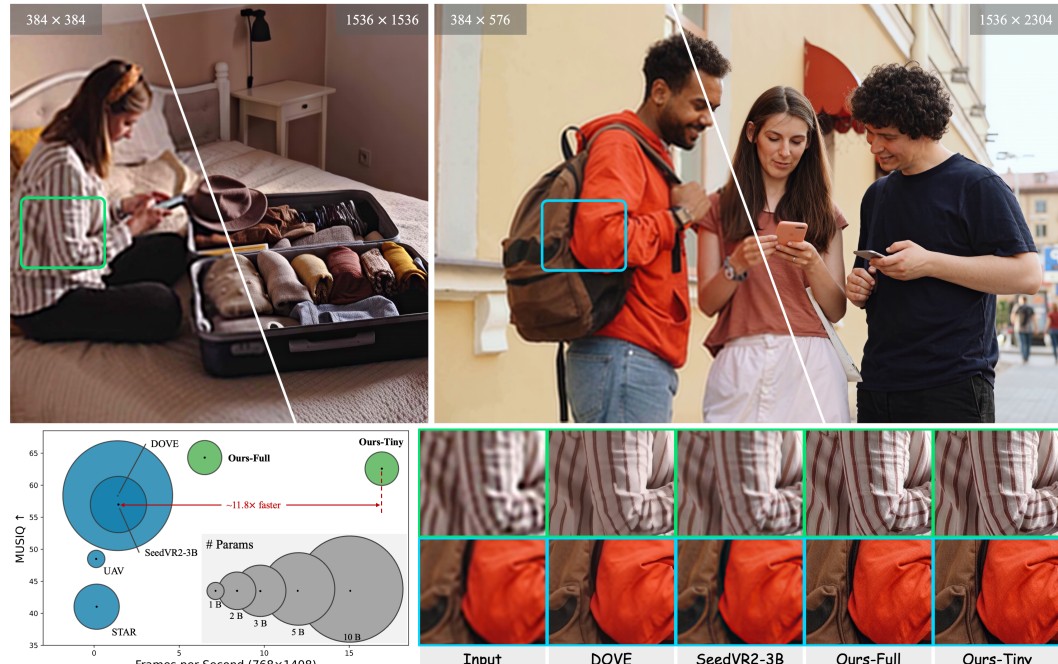

Figure 1: Efficiency and performance comparison in high-resolution video restoration. Compared to state-of-the-art VSR models (e.g., DOVE and SeedVR2-3B), FlashVSR restores sharper textures and more detailed structures. It achieves near real-time performance at 17 FPS on $768 \times 1408$ videos using a single A100 GPU, corresponding to a measured $11.8\times$ speedup over the fastest one-step diffusion VSR model. **(Zoom in for best view)**

et al., 2025) ($\sim 120\times$ faster) and the strongest one-step diffusion-based baseline SeedVR2-3B (Wang et al., 2025a) ($\sim 12\times$ faster). Additionally, benefiting from its streaming design we will soon introduce, FlashVSR only introduces 8 frames of lookahead latency, whereas previous chunk-based methods introduce latency equal to the segment length ($\sim 80$ frames). FlashVSR further scales reliably up to 1440p, delivering detail-rich videos, as shown in the right example of Fig. 1 and the Appendix C.1. FlashVSR achieves these advances through three key innovations.

**A training-friendly distillation scheme for one-step streaming VSR model.** Directly training an efficient and high-quality VSR model is challenging. Instead, we design a training-friendly three-stage processive distillation pipeline: (i) training a joint image–video VSR with full-attention VSR and using it as the teacher model, (ii) finetuning it to a block-sparse causal attention adaptation introduced above, and (iii) further distilling it into a one-step VSR model. Another advantage of FlashVSR is that it naturally supports parallel frame processing during training, since all latent frames only depend on the current LR frames as input. In contrast, in previous autoregressive VSRs (Huang et al., 2025; Lin et al., 2025b), the processing of a new frame only starts after the generation of the previous frame, resulting in heavy-loaded serial unfolding during training time.

**Sparse attention with locality constraints.** For video super-resolution, conventional dense 3D attention introduces substantial computational redundancy, which motivates us to adopt block-sparse attention for efficient processing. Specifically, we first compute a coarse attention map by pooling the key–value features, and then apply full attention only to the top-$k$ regions with the highest scores. Moreover, for high-resolution videos, we further introduce spatial local windows to constrain the attention range of each query, aligning the relative positional encoding range between training and inference. These designs improve the generalization of our model to high-resolution videos. To the best of our knowledge, this is the first diffusion-based VSR using sparse attention.

**Tiny conditional decoder.** After the accelerating diffusion transformer block (DiT), the causal 3D VAE decoder becomes the primary runtime bottleneck, consuming nearly 70% of the inference time at $768 \times 1408$ resolution. To address this, we introduce a tiny conditional decoder that leverages LR frames as auxiliary inputs in addition to latents, thereby simplifying HR reconstruction and enabling

a more compact design. With the same parameter budget, our ablation shows that it outperforms its unconditional variant, while preserving visual quality comparable to the original VAE decoder and reducing decoding time to roughly $1/7$.

Additionally, we construct a new large-scale dataset, **VSR-120K**, comprising 120k videos (average length >350 frames) and 180k high-quality images, filtered via automated quality control. Existing VSR datasets are limited in scale and video quality, *e.g.*, DOVE contains only 2K videos. Our dataset enables joint image-video training and will be publicly released to advance VSR research.

Extensive quantitative and qualitative results show that FlashVSR achieves state-of-the-art performance with up to $\sim 12\times$ speedup, while locality-constrained attention mitigates the resolution-induced train–test gap for robust generalization to ultra-high-resolution videos.

## 2 RELATED WORK

### 2.1 REAL-WORLD VIDEO SUPER-RESOLUTION

Early research in video super-resolution (VSR) often relied on simple synthetic degradations (*e.g.*, bicubic downsampling) (Wang et al., 2019; Fuoli et al., 2019; Li et al., 2020; Chan et al., 2021; 2022a; Cao et al., 2021; Liu et al., 2022a), but these methods perform poorly on real-world data. To mitigate this gap, some works collected paired data from consumer devices (Yang et al., 2021; Wei et al., 2023), but these datasets are small and sensor-biased. Later approaches introduced composite degradations combining blur, noise, and compression artifacts (Wang et al., 2021; Yang et al., 2021; Chan et al., 2022b; Zhou et al., 2024; Yang et al., 2024a; Xie et al., 2025).

The recent success of diffusion models in image restoration (Ai et al., 2024; Dong et al., 2025; Qu et al., 2024; Yu et al., 2024; Yue et al., 2025) has motivated their use in VSR (Zhou et al., 2024; Zhang & Yao, 2024; Yang et al., 2024a; Li et al., 2025; Xie et al., 2025). For instance, Upscale-A-Video (Zhou et al., 2024) employs optical-flow-guided propagation, MGLD-VSR (Yang et al., 2024a) introduces motion-aware objectives, and DiffVSR (Li et al., 2025) adopts staged optimization. More recent frameworks leverage large-scale video diffusion priors (Blattmann et al., 2023; Yang et al., 2024b; Wan et al., 2025), such as SeedVR (Wang et al., 2025b;a), and DOVE (Chen et al., 2025b). While effective in enforcing temporal consistency, these methods are hindered by the high cost of dense 3D attention and reliance on chunk-based inference, leading to large lookahead latency and inefficiency, even in one-step models.

### 2.2 STREAMING VIDEO DIFFUSION MODELS

Practical VSR must handle sequences lasting minutes or longer, making streaming capability essential for deployment. Recent studies have extended diffusion models from short clips to long streaming video generation. Diffusion Forcing (Chen et al., 2024a) reformulates denoising as block-wise sequential processing, enabling causal decoding. Follow-up works (Yin et al., 2024b; Chen et al., 2025a; Teng et al., 2025; Guo et al., 2025; Huang et al., 2025; Lin et al., 2025b) combined causal attention, KV-cache, and distillation to compress multi-step diffusion into a few denoising steps, enabling efficient online generation. However, these works primarily focus on video synthesis rather than restoration tasks such as VSR. Moreover, recent autoregressive video diffusion models rely on "student forcing" training to mitigate error accumulation, which requires predicting the previous output. This leads to serial unfolding of the forward process, reducing training efficiency.

### 2.3 VIDEO DIFFUSION ACCELERATION

Despite their strong performance, diffusion models remain computationally expensive, limiting real-time applications. Existing acceleration strategies mainly fall into three categories: **Feature caching.** Works such as DeepCache (Xu et al., 2018), FasterDiffusion (Li et al., 2023), and recent adaptations for video diffusion transformers(DiT) (Peebles & Xie, 2023; Selvaraju et al., 2024; Chen et al., 2024b; Liu et al., 2025) reduce redundancy by reusing intermediate activations. **One-step distillation.** Methods based on rectified flows (Liu et al., 2022b; 2023), score distillation (Lin et al., 2025a; Zhang et al., 2024), and adversarial training (Yin et al., 2024a; Wang et al., 2023b) compress iterative denoising into a single step, with representative examples including OSEDiff (Wu et al.,

2024), SinSR (Wang et al., 2024), and TSD-SR (Dong et al., 2025). Extensions to VSR such as DOVE (Chen et al., 2025b) and SeedVR2 (Wang et al., 2025a) achieve competitive results. **Sparse attention.** FlashAttention (Dao et al., 2022) improves memory and computation efficiency while also supporting block-sparse mechanisms, thereby reducing memory usage and accelerating attention computation. Techniques such as Sparse VideoGen (Xi et al., 2025), Sparse VideoGen2 (Yang et al., 2025), and others (Zhang et al., 2025; Shen et al., 2025) exploit spatiotemporal or semantic sparsity to reduce memory and computation while maintaining quality.

Despite recent advances, most methods remain inefficient, high-latency, and poorly generalized at high resolutions. In contrast, we introduce FlashVSR, the first diffusion-based VSR unifying one-step distillation, a train-friendly streaming design, and locality-constrained sparse attention. With a tiny conditional decoder for video restoration, FlashVSR addresses the key challenges of efficiency, temporal scalability, and high-resolution generalization, achieving near real-time performance and moving diffusion-based VSR closer to practical deployment.

## 3 METHOD

We present FlashVSR, an efficient diffusion-based one-step streaming framework for video super-resolution (VSR), achieving near real-time inference (17 FPS at $768 \times 1408$ on a single A100 GPU). Furthermore, to train a high-quality VSR, we also construct a large-scale high-quality dataset, VSR-120K. As illustrated in Fig. 2, FlashVSR builds on a three-stage distillation framework, enhanced with locality-constrained sparse attention to mitigate the train–infer resolution gap, and a tiny conditional decoder that reduces the cost of the 3D VAE decoder. Details are introduced below.

### 3.1 VSR-120K DATASET

To overcome the limited scale and quality of existing VSR datasets, we construct **VSR-120K**, a large-scale dataset for joint image–video super-resolution training. We collect raw data from open repositories such as Videvo, Pexels, and Pixabay[1], containing 600k video clips and 220k high-resolution images. For quality control, we employ LAION-Aesthetic predictor (Schuhmann et al., 2022) and MUSIQ (Ke et al., 2021) for visual quality, and RAFT (Teed & Deng, 2020) for motion filtering. The final dataset consists of 120k videos (average length >350 frames) and 180k high-quality images. We will release VSR-120K for public research use. See Appendix A for details.

### 3.2 THREE-STAGE DISTILLATION PIPELINE

To build a high-quality and efficient VSR model, we design a three-stage distillation pipeline: (1) joint video–image training to establish a strong teacher, (2) causal sparse attention adaptation for streaming efficiency, and (3) distribution-matching distillation into a one-step student.

**Stage 1. Video–Image Joint SR Training.** We adapt a pretrained video diffusion model (WAN2.1 1.3B (Wan et al., 2025)) to super-resolution by jointly training on videos and images, treating images as single-frame videos ($f{=}1$) to enable a unified 3D attention formulation. As illustrated in Stage 1 of Fig. 2, we apply a block-diagonal segment mask that restricts attention within the same segment:

$$\alpha_{ij} = \frac{\exp\!\left(\frac{q_i k_j^\top}{\sqrt{d}}\right) \mathbf{1}\!\left[\,\mathrm{seg}(i) = \mathrm{seg}(j)\,\right]}{\sum_l \exp\!\left(\frac{q_i k_l^\top}{\sqrt{d}}\right) \mathbf{1}\!\left[\,\mathrm{seg}(i) = \mathrm{seg}(l)\,\right]}, \tag{1}$$

where $\mathrm{seg}(i)$ denotes the segment identity of token $i$ (image or video clip) and $\alpha$ denotes the normalized attention weight. Block-sparse constraints are omitted so the teacher retains full spatiotemporal priors. A fixed text prompt is used for conditioning, with cross-attention keys and values reused across samples. We further introduce a lightweight low-resolution(LR) Proj-In layer to project LR inputs into the feature space instead of using the VAE encoder. Training employs the standard flow matching loss (Lipman et al., 2022).

**Stage 2. Block-Sparse Causal Attention Adaptation.** We adapt the Stage 1 full-attention DiT into a Sparse-Causal DiT by introducing causal masking and block-sparse attention, as illustrated

---

[1]https://www.videvo.net/, https://www.pexels.com/, https://pixabay.com/

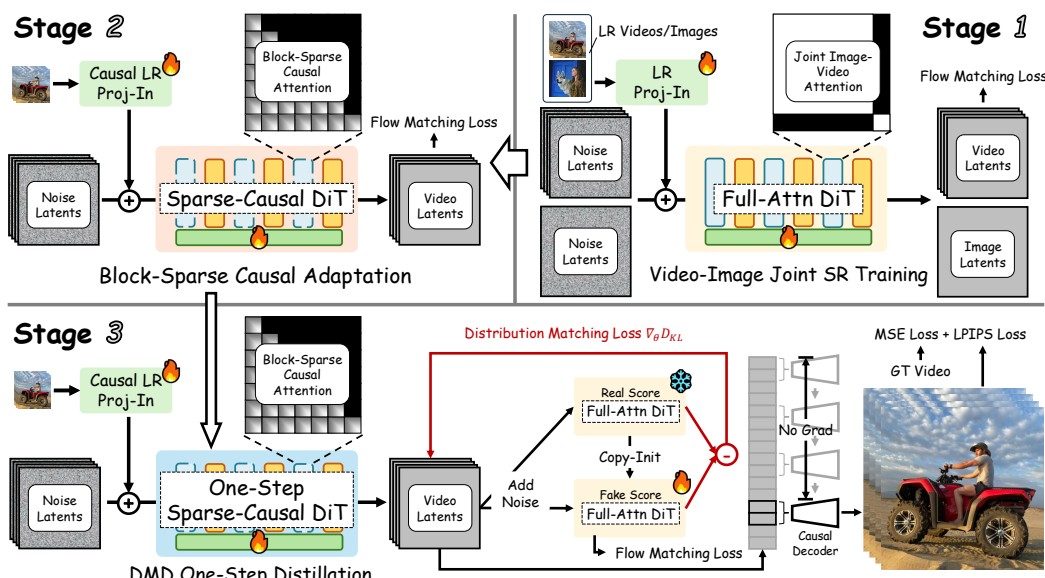

Figure 2: Overview of the three-stage training pipeline of FlashVSR, covering video–image joint SR training, adaptation with block-sparse causal attention for streaming inference, and distribution-matching one-step distillation combined with reconstruction supervision.

in Fig. 2. The causal mask restricts each latent to its current and past positions. Following (Zhang et al., 2025; Shen et al., 2025), queries and keys are partitioned into non-overlapping blocks of size $(2, 8, 8)$ and reshaped into $(B, block\_num, 128, C)$ with $block\_num = L/128$. Within each block, average pooling yields compact block-level features to compute a coarse block-to-block attention map. The top-$k$ most relevant block pairs are selected, and full $128 \times 128$ attention is applied only to these regions with the original $(Q, K, V)$. This reduces attention cost to 10–20% of the dense baseline without performance loss. The LR Proj-In layer is converted to a causal variant for streaming inference, and training continues with the flow matching loss on video data only.

**Stage 3. Distribution-Matching One-Step Distillation.** Recent studies on one-step streaming video diffusion focus on video generation, typically requiring clean past frames as input for motion plausibility. Teacher forcing (conditioning on ground truth) (Lin et al., 2025b) causes error accumulation during inference, while student forcing (conditioning on predicted latents) (Lin et al., 2025b; Huang et al., 2025) alleviates this but requires sequential unfolding, reducing efficiency.

In Stage 3 (Fig. 2), we refine the Stage 2 sparse-causal DiT into a one-step model $G_{one}$ and propose a parallel training paradigm for streaming VSR. The model takes LR frames and Gaussian noise as input, with all latents trained under a unified timestep using a block-sparse causal attention mask. The Stage 1 full-attention DiT serves as the teacher $G_{real}$, while its copy $G_{fake}$ learns the distribution of fake latents, following the DMD pipeline by Yin et al. (2024a)). Here, $z_{pred}$ denotes the predicted latent, and $x_{pred}$ the reconstructed HR frame. The overall objective combines distribution-matching distillation, flow matching, and pixel-space reconstruction losses:

$$\mathcal{L} = \underbrace{\mathcal{L}_{DMD}(z_{pred}, G_{one}, G_{real}, G_{fake})}_{\text{distribution-matching distillation}} + \underbrace{\mathcal{L}_{FM}(z_{pred}, G_{fake})}_{\text{flow matching}} + \underbrace{\|x_{pred} - x_{gt}\|_2^2 + \lambda\,\mathcal{L}_{lpips}(x_{pred}, x_{gt})}_{\text{decoder reconstruction}}, \quad (2)$$

where $\lambda = 2$. Due to memory constraints, two latents are randomly selected per iteration for decoding, with previous ones detached from gradients.

Since training and inference rely only on LR frames and noise, the train–infer gap is eliminated. As a one-step model, the later layers of $G_{one}$ already contain clean latent information propagated via the KV-cache for temporal continuity. *The core insight is that*, unlike video generation, VSR is strongly conditioned on LR frames, so clean historical latents are unnecessary for motion plausibility. The model focuses on content reconstruction, while temporal consistency is refined in later layers via the KV-cache. This design enables efficient parallel training while preserving high fidelity and eliminating the train–infer gap. Further discussion is provided in Appendix B.4.

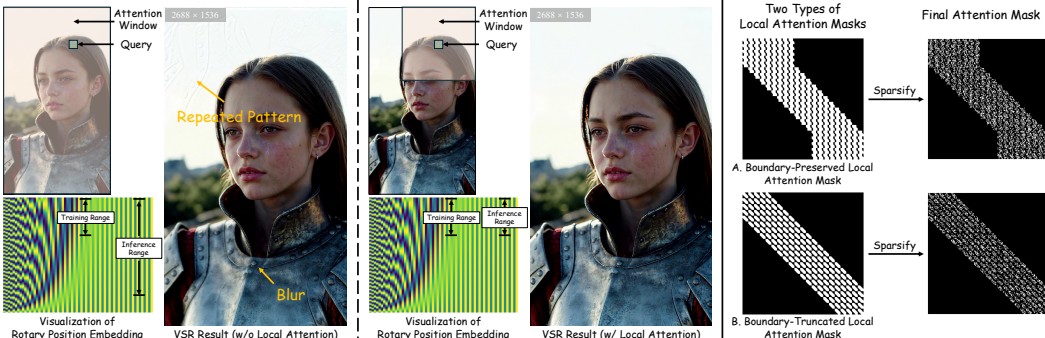

Figure 3: Locality-Constrained Sparse Attention. Left: At ultra-high resolutions, performing inference beyond the trained positional encoding range produces artifacts (e.g., repetition or blur). Restricting each query to a local attention window keeps the positional encoding range consistent between training and inference, thereby preventing artifacts. Right: Two local window rules, namely boundary-preserved and boundary-truncated, are illustrated. The final sparse attention mask is computed within these local masks.

### 3.3 Locality-Constrained Sparse Attention

We observed that for super-resolution, model trained medium-resolution may fail to generalize to ultra-high resolutions (e.g., 1440p), leading to repeated patterns and blurring as shown in illustrated in Fig. 3. Our analysis shows that this arises from the periodicity of positional encodings: when the positional range at inference far exceeds that seen during training, certain dimensions repeat their patterns, degrading self-attention, as shown in the bottom of Fig. 3.

To address this, we introduce **locality-constrained attention**, which restricts each query at inference to a *limited spatial neighborhood*, thereby aligning the attention range with that used in training. With the relative formulation of RoPE (Su et al., 2024), this simple constraint eliminates the train–inference gap of positional range. This approach closes the resolution gap and enables consistent performance even on high-resolution inputs, as shown in the middle of Fig. 3. This design is further validated through quantitative results in Sec. 4.4 and qualitative visualizations in Appendix C.1.

### 3.4 Tiny Conditional Decoder

After obtaining the one-step streaming model, we find the VAE decoder dominates inference ($\sim 70\%$ runtime) and becomes the bottleneck. To address this, we design a **Tiny Conditional Decoder** (TC Decoder, Fig. 4), which, instead of merely shrinking the original VAE decoder, conditions reconstruction on both LR frames and latents. This reduces decoding complexity and preserves fine details with fewer parameters. Let $x_{\text{pred}}$ denote the reconstructed HR

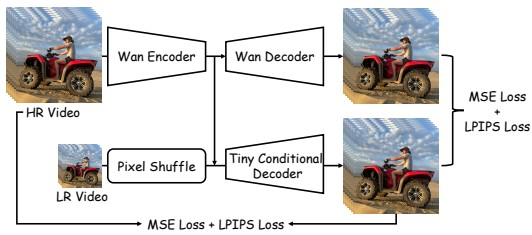

Figure 4: Training pipeline of the TC Decoder.

frame, $x_{\text{gt}}$ the ground truth, and $x_{\text{wan}}$ the Wan decoder output. Training combines pixel-level supervision with distillation from the original Wan decoder:

$$\mathcal{L} = \|x_{\text{pred}} - x_{\text{gt}}\|_2^2 + \lambda \mathcal{L}_{\text{LPIPS}}(x_{\text{pred}}, x_{\text{gt}}) + \|x_{\text{pred}} - x_{\text{wan}}\|_2^2 + \lambda \mathcal{L}_{\text{LPIPS}}(x_{\text{pred}}, x_{\text{wan}}), \qquad (3)$$

where $\lambda = 2$. The TC Decoder achieves nearly $7\times$ faster decoding than the original VAE decoder while maintaining comparable quality, and under the same parameter budget it consistently outperforms unconditional tiny decoders.

## 4 Experiments

### 4.1 Implementation Details

FlashVSR is built upon Wan 2.1–1.3B (Wan et al., 2025) and fine-tuned with LoRA (Hu et al., 2022) (rank 384). All stages are trained on the VSR-120K dataset, with paired LR–HR videos and images

Table 1: Quantitative comparison on YouHQ40, REDS, SPMCS (synthetic), VideoLQ (real) and AIGC30 (AIGC). Best in **red**, second-best in blue.

| Dataset | Metric | Upscale-A-Video | STAR | RealViformer | DOVE | SeedVR2-3B | Ours-Full | Ours-Tiny |
|---------|--------|-----------------|------|--------------|------|------------|-----------|-----------|
| **YouHQ40** | PSNR ↑ | 23.19 | 23.19 | 23.67 | **24.39** | 23.05 | 23.13 | 23.31 |
| | SSIM ↑ | 0.6075 | 0.6388 | 0.6189 | **0.6651** | 0.6248 | 0.6004 | 0.6110 |
| | LPIPS ↓ | 0.4585 | 0.4705 | 0.4476 | 0.4011 | 0.3876 | 0.3874 | **0.3866** |
| | NIQE ↓ | 4.834 | 7.275 | **3.360** | 4.890 | 3.751 | 3.382 | 3.489 |
| | MUSIQ ↑ | 43.07 | 35.05 | 62.73 | 61.60 | 62.31 | **69.16** | 66.63 |
| | CLIPIQA ↑ | 0.3380 | 0.2974 | 0.4451 | 0.4437 | 0.4909 | **0.5873** | 0.5221 |
| | DOVER ↑ | 6.889 | 7.363 | 9.739 | 11.29 | 12.43 | **12.71** | 12.66 |
| **REDS** | PSNR ↑ | 24.84 | 24.01 | **25.96** | 25.60 | 24.83 | 23.92 | 24.11 |
| | SSIM ↑ | 0.6437 | 0.6765 | 0.7092 | **0.7257** | 0.7042 | 0.6491 | 0.6511 |
| | LPIPS ↓ | 0.4168 | 0.371 | **0.2997** | 0.3077 | 0.3124 | 0.3439 | 0.3432 |
| | NIQE ↓ | 3.104 | 4.776 | 2.722 | 3.564 | 3.066 | **2.425** | 2.680 |
| | MUSIQ ↑ | 53.00 | 46.25 | 63.23 | 65.51 | 61.83 | **68.97** | 67.43 |
| | CLIPIQA ↑ | 0.2998 | 0.2807 | 0.3583 | 0.4160 | 0.3695 | **0.4661** | 0.4215 |
| | DOVER ↑ | 6.366 | 6.309 | 8.338 | **9.368** | 8.725 | 8.734 | 8.665 |
| **SPMCS** | PSNR ↑ | 23.95 | 23.68 | **25.61** | 25.46 | 23.62 | 23.84 | 24.02 |
| | SSIM ↑ | 0.6209 | 0.6700 | 0.7030 | **0.7201** | 0.6632 | 0.6346 | 0.6450 |
| | LPIPS ↓ | 0.4277 | 0.3910 | 0.3437 | **0.3289** | 0.3417 | 0.3436 | 0.3451 |
| | NIQE ↓ | 3.818 | 7.049 | 3.369 | 4.168 | 3.425 | **3.151** | 3.302 |
| | MUSIQ ↑ | 54.33 | 45.03 | 65.32 | 69.08 | 66.87 | **71.05** | 69.77 |
| | CLIPIQA ↑ | 0.4060 | 0.3779 | 0.4150 | 0.5125 | 0.5307 | **0.5792** | 0.5238 |
| | DOVER ↑ | 5.850 | 4.589 | 8.083 | **9.525** | 8.856 | 9.456 | 9.426 |
| **VideoLQ** | NIQE ↓ | 4.889 | 5.534 | **3.428** | 5.292 | 5.205 | 3.803 | 4.070 |
| | MUSIQ ↑ | 44.19 | 40.19 | **57.60** | 45.05 | 43.39 | 55.48 | 52.27 |
| | CLIPIQA ↑ | 0.2491 | 0.2786 | 0.3183 | 0.2906 | 0.2593 | **0.4184** | 0.3601 |
| | DOVER ↑ | 5.912 | 5.889 | 6.591 | 6.786 | 6.040 | **8.149** | 7.481 |
| **AIGC30** | NIQE ↓ | 5.563 | 6.212 | 4.189 | 4.862 | 4.271 | **3.871** | 4.039 |
| | MUSIQ ↑ | 47.87 | 38.62 | 50.74 | 50.59 | 50.53 | **56.89** | 55.80 |
| | CLIPIQA ↑ | 0.4317 | 0.3593 | 0.4510 | 0.4665 | 0.4767 | **0.5543** | 0.5087 |
| | DOVER ↑ | 10.24 | 11.00 | 11.24 | 12.34 | 12.48 | **12.65** | 12.50 |

synthesized via the RealBasicVSR degradation pipeline (Chan et al., 2022b). Training is conducted on 32 A100-80G GPUs, while evaluation uses a single A100. All stages use a batch size of 32, taking about 2, 1, and 2 days for Stages 1–3, respectively. Stage 1 uses 89-frame clips ($768 \times 1280$) and paired images; Stage 2 continues training with videos only; Stage 3 adopts the same setting. The AdamW optimizer (Loshchilov & Hutter, 2017) is used with learning rate $1 \times 10^{-5}$ and weight decay 0.01. The TC Decoder is trained separately on 61-frame clips ($384 \times 384$) for about 2 days.

## 4.2 DATASETS, METRICS, AND BASELINES

We evaluate on three synthetic datasets (YouHQ40 (Zhou et al., 2024), REDS (Nah et al., 2019), SPMCS (Yi et al., 2019)), one real-world dataset (VideoLQ (Chan et al., 2022b)), and an AI-generated set (AIGC30). Synthetic LR frames are generated using the same degradation pipeline as training. We evaluate with PSNR (Wikipedia contributors, 2024), SSIM (Wang et al., 2004), LPIPS (Zhang et al., 2018), MUSIQ (Ke et al., 2021), CLIPIQA (Wang et al., 2023a), and DOVER (Wu et al., 2023) on datasets with ground truth (YouHQ40, REDS, SPMCS), and only with the no-reference metrics (MUSIQ, CLIPIQA, DOVER) on datasets without ground truth (VideoLQ, AIGC30). We compare FlashVSR against RealViFormer (Zhang & Yao, 2024) (non-diffusion transformer), STAR (Xie et al., 2025) and Upscale-A-Video (Zhou et al., 2024) (multi-step diffusion), and DOVE (Chen et al., 2025b) and SeedVR2-3B (Wang et al., 2025a) (one-step diffusion). We also report results using the Wan Decoder (Ours-Full) and the proposed TC Decoder (Ours-Tiny).

## 4.3 COMPARISON WITH EXISTING METHODS

**Quantitative Comparisons.** We compare FlashVSR with state-of-the-art real-world video super-resolution methods. For multi-step diffusion-based models, we adopt their default configurations,

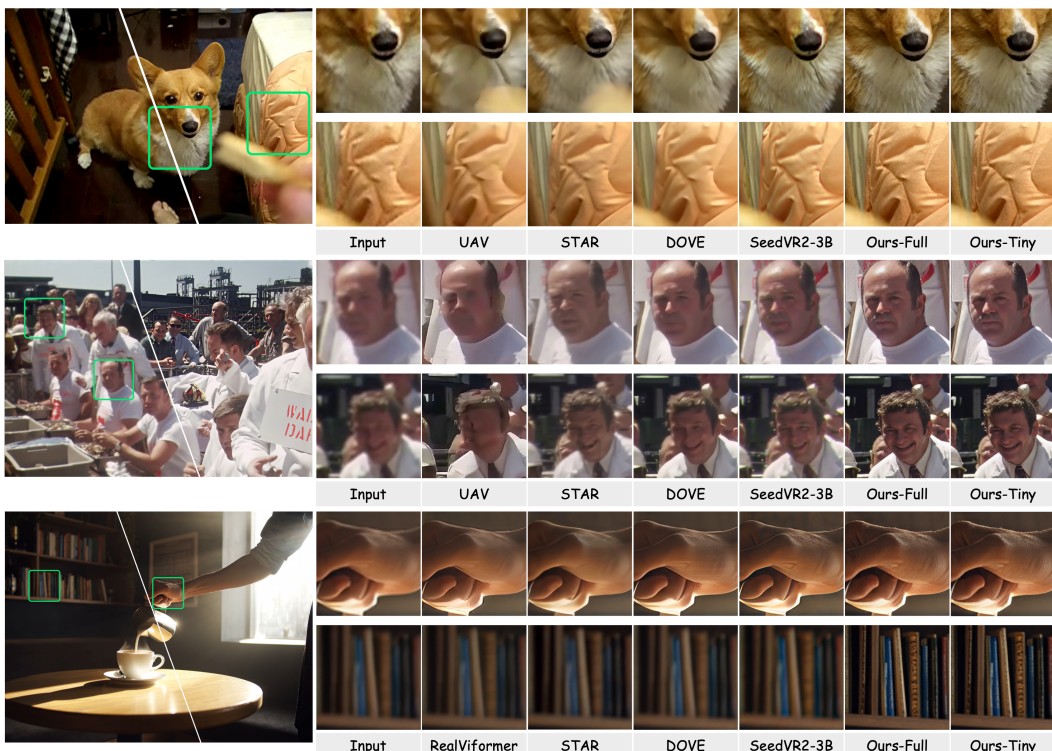

Figure 5: Visualization results of video super-resolution on real-world and AIGC videos.

Table 2: Efficiency comparison (peak memory, runtime, and parameters) on $101 \times 768 \times 1408$ videos. FPS is computed from the measured runtime.

| Metric | Upscale-A-Video | STAR | DOVE | SeedVR2-3B | Ours-Full | Ours-Tiny |
|---|---|---|---|---|---|---|
| Peak Mem. (GB) | 18.39 | 24.86 | 25.44 | 52.88 | 18.33 | 11.13 |
| Runtime (s) / FPS | 811.71 / 0.12 | 682.48 / 0.15 | 72.76 / 1.39 | 70.58 / 1.43 | 15.50 / 6.52 | 5.97 / 16.92 |
| Params (M) | 1086.75 | 2492.90 | 10548.57 | 3391.48 | 1780.14 | 1752.18 |

using 15 sampling steps for STAR and 30 steps for Upscale-A-Video. Tab. 1 reports the quantitative results. FlashVSR consistently outperforms competing methods across all datasets, achieving superior performance particularly on perceptual metrics such as MUSIQ, CLIPIQA, and DOVER. Moreover, compared to using the original VAE decoder from Wan, the proposed TC Decoder further improves reconstruction metrics while maintaining efficiency. We also note that RealViFormer has an inherent advantage on REDS, since this dataset is included in its training set. Overall, the evaluation highlights the effectiveness of FlashVSR in delivering high-quality video super-resolution.

**Qualitative Comparisons.** To provide a more intuitive comparison of visual quality in real-world scenarios, we present qualitative results on VideoLQ and AIGC30 in Fig. 5. For clarity, we also enlarge selected local patches to better illustrate the differences among the LR frames and outputs of all methods. FlashVSR produces sharper and more detailed reconstructions compared to baselines, with textures and structures that appear more natural. For instance, in the last row of Fig. 5, FlashVSR restores clearer hand textures and bookshelf details, yielding results that are visually more realistic. These qualitative observations are consistent with the quantitative improvements on perceptual metrics. Additional visualizations are provided in the Appendix C.1.

**Efficiency Analysis.** Tab. 2 reports the efficiency comparison on 101-frame videos at $768 \times 1408$ resolution. With streaming inference, block-sparse attention, one-step distillation, and a lightweight conditional decoder, FlashVSR achieves substantial efficiency gains over all baselines. It runs $136\times$ faster than Upscale-A-Video (30 steps), $114\times$ faster than STAR (15 steps), and still $11.8\times$ faster than the fastest one-step model SeedVR2-3B, while also using much less peak memory (11.1 GB

Table 5: Ablation of Locality-constrained Attention. Both Boundary-Truncated and Boundary-Preserved outperform Global Attention across all metrics. Best in **red**, second-best in blue.

| Method | PSNR ↑ | SSIM ↑ | LPIPS ↓ | NIQE ↓ | MUSIQ ↑ | CLIPIQA ↑ | DOVER ↑ |
|---|---|---|---|---|---|---|---|
| Global Attention | 24.21 | 0.6988 | 0.3423 | 3.152 | 65.57 | 0.5594 | 9.1259 |
| Boundary-Truncated | 24.60 | 0.7155 | 0.3385 | **2.850** | **67.47** | **0.6278** | **9.5132** |
| Boundary-Preserved | **24.87** | **0.7232** | **0.3304** | 2.968 | 67.16 | 0.6029 | 9.3259 |

vs. 52.9 GB). STAR uses chunk-wise inference (chunk size 32, overlap 0.5), and most methods process entire sequences in one pass. In contrast, FlashVSR adopts streaming inference, reducing lookahead latency to just 8 frames (vs. 32 for STAR and 101 for others). These results demonstrate the practicality of FlashVSR for real-world deployment.

## 4.4 ABLATION STUDY

**Sparse Attention.** We evaluate the impact of sparse attention on REDS. As shown in Tab. 3, FlashVSR with 13.6% sparsity achieves nearly identical reconstruction and perceptual quality compared to a full-attention baseline (KV-cache size 85 frames). At $768 \times 1408$, it reduces inference time from 1.105s to 0.355s per 8 frames ($3.1\times$ speedup), thereby substantially improving efficiency without compromising visual quality. This indicates that sparse attention effectively prunes redundant interactions, mitigating computational overhead while preserving essential spatiotemporal dependencies for high-quality video super-resolution.

Table 3: Sparse vs. Full Attention.

| Metric | 13.6% Sparse | Full Attn. |
|---|---|---|
| PSNR ↑ | 24.11 | 24.65 |
| SSIM ↑ | 0.6511 | 0.6630 |
| LPIPS ↓ | 0.3432 | 0.3320 |
| NIQE ↓ | 2.680 | 2.878 |
| MUSIQ ↑ | 67.43 | 65.77 |
| CLIPIQA ↑ | 0.4215 | 0.4110 |
| DOVER ↑ | 8.665 | 8.750 |

**Tiny Conditional Decoder.** We evaluate the proposed TC Decoder on 200 randomly selected unseen videos, where all inputs are compressed using the Wan VAE encoder and reconstructed by three decoders: the original Wan decoder, our TC Decoder, and an unconditional variant. As shown in Tab. 4 and Fig. 5, the TC

Table 4: Ablation of tiny conditional decoder.

| Metric | Wan Decoder | Ours | Unconditional Variant |
|---|---|---|---|
| PSNR ↑ | 32.58 | 31.08 | 29.96 |
| SSIM ↑ | 0.9417 | 0.9244 | 0.9079 |
| LPIPS ↓ | 0.0715 | 0.1014 | 0.1231 |

Decoder achieves nearly identical visual quality to the Wan decoder, and its quantitative metrics are also close. For a 101-frame video at $768 \times 1408$ resolution, it decodes in 1.60s compared to 11.13s for the Wan decoder, achieving a $\sim 7\times$ speedup. Moreover, it consistently outperforms the unconditional variant across PSNR, SSIM, and LPIPS, demonstrating the effectiveness of incorporating LR frame conditions. The TC Decoder provides substantial acceleration with minimal fidelity loss, making it well-suited for practical video super-resolution deployment.

**Locality-constrained Attention.** Fig. 3 illustrates that the proposed locality-constrained attention mask alleviates repeated patterns and blurring at ultra-high resolutions by aligning positional encoding ranges between training and inference. To quantitatively validate its effectiveness, we evaluate on 15 high-resolution videos ($1536 \times 2688$, average 305 frames). We consider two variants depending on boundary handling (Fig. 3): Boundary-Preserved and Boundary-Truncated, both with a receptive field restricted to $1152 \times 1152$ and matched sparsity to global attention. Results are reported in Tab. 5. Compared to global attention, both variants consistently improve across all metrics. Notably, Boundary-Truncated achieves slightly higher perceptual quality, while Boundary-Preserved maintains competitive performance with better fidelity. These results confirm that locality-constrained attention effectively enhances VSR performance on ultra-high-resolution videos.

## 5 CONCLUSION

We presented FlashVSR, an efficient diffusion-based one-step framework for streaming video super-resolution. By combining streaming distillation, locality-constrained sparse attention, and a tiny conditional decoder, FlashVSR achieves state-of-the-art quality with near real-time efficiency and strong scalability to ultra-high resolutions. Our results demonstrate both its effectiveness and practicality, underscoring the potential of FlashVSR for real-world video applications.

## REPRODUCIBILITY STATEMENT

We will release the code, the VSR-120K dataset, and pretrained model weights to facilitate reproducibility. Implementation details are provided in Sec. 4.1 and Appendix B, while the evaluation benchmarks and metrics are described in Sec. 4.2.

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

## A  VSR-120K Dataset

### A.1  Data Sources.

The VSR-120K dataset is constructed from publicly accessible video and image platforms, including Videvo, Pexels, and Pixabay. We begin with approximately 600k raw video clips. To ensure sufficient spatial resolution, only videos with resolution higher than 1080p are retained. These videos, primarily professional stock footage, generally exhibit higher visual quality than typical web-crawled content, while also providing diverse temporal dynamics essential for video super-resolution. In addition to video data, we gather 220k high-resolution photos from Pexels, requiring the shorter side to exceed 1024 pixels. A substantial portion of these images are 4K or higher, offering clean textures and fine-grained spatial details. Since videos are usually stored in compressed formats (e.g., mp4), individual frames may suffer from compression artifacts or degraded quality. The image subset therefore complements the video corpus by providing pristine high-fidelity supervision for spatial reconstruction. After filtering, the combined video–image corpus provides a large-scale, high-quality foundation for joint training. Videos contribute both strong visual quality and diverse temporal information, while images supply superior spatial detail to mitigate compression artifacts. Together, they form a complementary dataset that supports robust VSR model development.

### A.2  Visual Quality Filtering.

Although stock photography and video generally exhibit high visual quality, many samples still suffer from extensive defocus regions, low contrast, or other artifacts that make them unsuitable for VSR training. To automatically filter such cases, we adopt two complementary predictors. The LAION-Aesthetic predictor provides a single global score, while MUSIQ evaluates both the full image and four local crops (top-left, top-right, bottom-left, bottom-right). For each image, we take the aesthetic score and the averaged MUSIQ scores as its final quality indicators. For videos, each clip is divided into 8-second segments, from which 4 frames are uniformly sampled. Each sampled frame is evaluated with the same protocol, and the average score across the 4 frames is assigned to the segment. Segments below the threshold are discarded, while consecutive high-quality segments are merged into a single annotation with start and end frame indices.

### A.3  Motion Filtering.

For video super-resolution, static or near-static clips provide limited temporal information. To guarantee sufficient motion diversity, we estimate optical flow using RAFT (Teed & Deng, 2020) on the sampled frames of each segment. The $\ell_2$ norm of the flow field is used to quantify motion strength. Segments with insufficient average motion are filtered out.

After the above filtering steps, we obtain 120k high-quality video clips and 180k high-resolution images from the initial 600k videos and 220k images. The primary goal of VSR-120K is to provide a large-scale resource for robust training. This dataset supports joint image–video super-resolution learning and offers a reliable foundation for training FlashVSR as well as future VSR models.

## B  Implementation Details

### B.1  Causal LR Projection-In Layer

In streaming VSR, it is crucial to efficiently incorporate information from low-resolution (LR) frames into the diffusion backbone while preserving temporal causality. To this end, we propose a lightweight **Causal LR Projection-In Layer**, as shown in Fig. 6. The design aligns with Wan's Video DiT compression scheme, which applies $4\times$ temporal compression and $8\times$ spatial compression. Specifically, every 4 consecutive LR frames are grouped into a video clip. Each clip is first passed through a 2D pixel-shuffle layer to match the $8\times$ spatial compression. The resulting feature map is then processed by two 3D CausalConv layers, each performing $2\times$ temporal compression, to produce a compact representation of the current clip. To ensure temporal consistency, we maintain a causal cache: the feature representation from the previous clip is carried forward and fused with the current one, ensuring smooth propagation of conditional information across time while maintaining

Causal LR Projection-In Layer

Figure 6: Architecture of the Causal LR Projection-In Layer.

Table 6: Quantitative results of different KV-cache eviction strategies on the REDS dataset.

| Metric | Sliding-Window | Uniform Eviction | Head-wise Eviction |
|---|---|---|---|
| PSNR ↑ | 24.11 | 24.31 | 23.61 |
| SSIM ↑ | 0.6511 | 0.6601 | 0.6461 |
| LPIPS ↓ | 0.3432 | 0.3439 | 0.3812 |
| NIQE ↓ | 2.680 | 2.770 | 3.107 |
| MUSIQ ↑ | 67.43 | 66.12 | 62.71 |
| CLIPIQA ↑ | 0.4215 | 0.4097 | 0.3615 |
| DOVER ↑ | 8.665 | 8.411 | 8.021 |

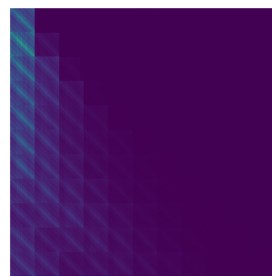

Figure 7: Illustration of the sink attention effect in specific attention heads.

causality. Finally, an MLP layer aggregates the compressed features and projects them into the same latent space as the DiT backbone. The resulting conditional embedding is added element-wise to the patchified latent tokens of the DiT, allowing the LR frames to directly guide high-resolution reconstruction. This design offers two key advantages: (i) it introduces LR guidance with negligible overhead, and (ii) the causal cache mechanism ensures consistency across clips during streaming inference.

## B.2 FIXED TEXT CONDITIONING

In practical streaming VSR tasks, video content may vary significantly, and providing a scene-specific caption for each case would introduce additional computational overhead. To avoid this issue, we adopt a fixed prompt as the textual condition for all scenes: *"Cinematic, High Contrast, highly detailed, taken using a Canon EOS R camera, hyper detailed photo-realistic maximum detail, Color Grading, ultra HD, extreme meticulous detailing, skin pore detailing, hyper sharpness, perfect without deformations."* This strategy not only eliminates the cost of generating dynamic captions but also simplifies the image–video joint training process, since both images and videos can share the same cross-attention key–value representations.

## B.3 KV-CACHE EVICTION STRATEGIES

By default, FlashVSR adopts a sliding-window strategy for KV-cache eviction. However, we also conduct a preliminary exploration of alternative strategies, such as importance-based eviction. The importance score is derived from the coarse block-to-block attention map of the previous latent, and the least important KV entries are discarded to retain only the most relevant context. We evaluate two variants on the REDS dataset: (i) **head-wise eviction**, where KV entries are evicted independently for each attention head within a layer, and (ii) **uniform eviction**, where all heads in the same attention layer share a common eviction decision. Results are reported in Table 6. The uniform evic-

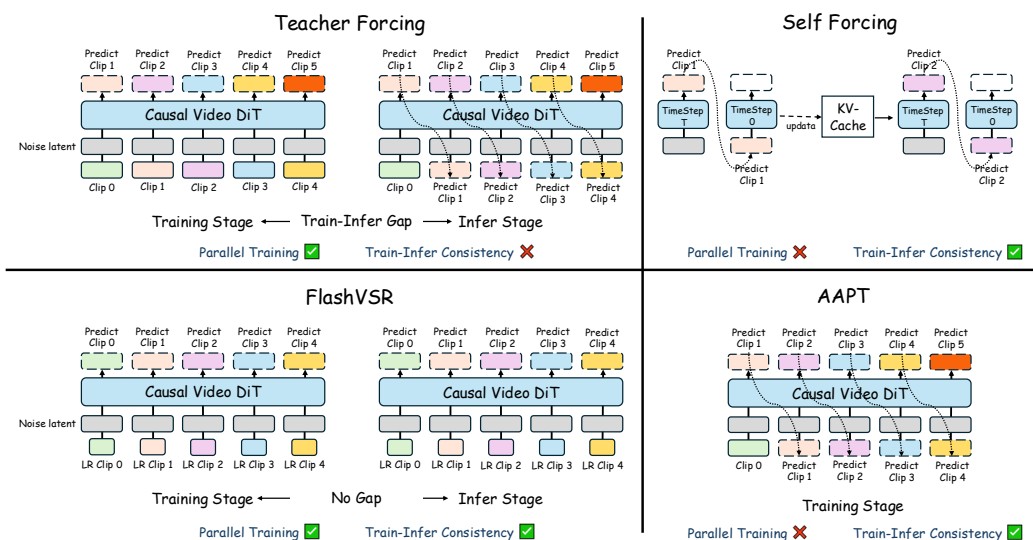

Figure 8: Comparison of four training pipelines for stream video diffusion: Teacher Forcing, AAPT, Self-Forcing, and FlashVSR.

tion strategy provides no improvement over the sliding-window baseline, while head-wise eviction leads to a clear performance drop.

We attribute this degradation to the lack of temporal continuity in FlashVSR's attention maps: importance scores from one latent do not reliably transfer to the next. Furthermore, we observe a *sink attention* effect in certain heads, where attention is disproportionately focused on the first frame (Fig. 7). Under head-wise eviction, this bias causes the KV entries of the first frame to be consistently preserved, leading to blurred textures and structural distortions.

### B.4 DISCUSSION ON STREAM VSR PIPELINES

In Fig. 8, we illustrate four training *pipelines* for stream video diffusion models: Teacher Forcing (Lin et al., 2025b), AAPT (Lin et al., 2025b), Self-Forcing (Huang et al., 2025), and our proposed **FlashVSR**. The first three are devised for *video generation*, whereas FlashVSR targets *video super-resolution (VSR)*.

**Teacher Forcing.** During training, Teacher Forcing conditions the model on the ground-truth previous video clip, whereas inference relies on the previously predicted clip. This mismatch introduces a train–inference gap, leading to error accumulation during inference (Lin et al., 2025b).

**AAPT.** AAPT unfolds the forward pass sequentially, conditioning each predicted clip on the previously generated one. This setup aligns the training and inference procedures but prevents parallel training, limiting efficiency.

**Self Forcing.** Similar to AAPT, Self Forcing also requires waiting for the output of the previous predicted clip. At timestep 0, this output is fed back into the Video DiT, updating the KV-cache. The updated cache then guides the generation of the current video clip, ensuring consistency with the inference procedure. However, like AAPT, this approach also prohibits parallel training.

We posit that the above dilemma stems from the objective of *video generation*, which must model motion plausibility from scratch and thus benefits from explicitly accessing complete past predictions early in the stack. In contrast, *VSR* receives strong conditioning from the low-resolution (LR) frames; the model's main burden is spatial detail restoration rather than free-form motion synthesis. In a one-step formulation, later layers naturally refine *clean* latent features that summarize historical context useful for temporal alignment. **FlashVSR.** Guided by this insight, FlashVSR discards past predicted clips as inputs. As shown in Fig. 8, both training and inference take only LR frames and noise latents. This eliminates the train–inference gap while enabling efficient parallel training.

At inference, FlashVSR runs causally in a streaming manner: for frame $t$, the model consumes the current LR frame and noise $\epsilon_t$ and produces the latent in a single step,

$$z_t = G_{\text{one}}(\text{LR}_t, \epsilon_t; \text{KV}_{<t}),$$

where $\text{KV}_{<t}$ stores the cached keys and values from the most recent latents within a sliding window across all layers. After generating $z_t$, the cache is updated and carried forward to the next frame.

*How temporal consistency is ensured.* The cached representations from different layers play complementary roles. Early-layer KV states aggregate contextual cues primarily aligned with the LR inputs, ensuring that structural and motion information remains faithful to the observed low-resolution video. Later-layer KV states contain progressively cleaner latent representations, which encode refined high-frequency details. When propagated across frames, these cleaner features stabilize textures and fine details over time. By jointly leveraging early- and late-layer memory, the KV-cache implicitly aligns content across frames, ensuring temporal consistency, even though FlashVSR conditions only on the current LR inputs.

## C  More Experiment Results

### C.1  Additional Visualization Results

To further demonstrate the effectiveness of our approach in practical scenarios, we provide additional visualizations on both real-world video super-resolution and AIGC video enhancement, as shown in Fig. 9. It can be observed that our method achieves clearer frames with superior detail restoration compared to existing baselines.

In addition, Fig. 10 shows qualitative results on high-resolution videos ($1536 \times 2688$), highlighting the effectiveness of our proposed locality-constrained attention. While global attention suffers from repeated textures and blurring due to mismatched positional encoding ranges, both Boundary-Preserved and Boundary-Truncated variants produce sharper and more stable frames by aligning positional encoding ranges between training and inference. These visual findings are consistent with the quantitative analysis in Sec. 4.4.

We also provide a demo video `[FlashVSR.mp4]` to further showcase the super-resolution capability of FlashVSR. Due to file size limitations, the demo video may be compressed; the original version exhibits even higher visual quality.

### C.2  User Study

We conducted a blind user study to compare five one-step VSR models: Ours-Tiny, Ours-Full, SeedVR2-3B, DOVE, and RealViformer, based on human preference. Following APT Lin et al. (2025a), we employed the GSB test to compute preference scores. Specifically, the score is defined as $\frac{G-B}{G+S+B}$, where $G$ is the number of samples judged as "good," $B$ is the number of samples judged as "bad," and $S$ is the number of samples considered "similar" (*i.e.*, no preference). The score ranges from $-100\%$ to $100\%$, with $0\%$ indicating no preference between the compared methods. The study includes 32 test sets, covering both real-world and AIGC-degraded low-resolution videos. Participants evaluated three aspects: **Overall Quality**, **Video Fidelity**, and **Video Quality**. Twenty researchers with backgrounds in computer vision participated in the user study. The user study interface is shown in Fig. 11, and the results are summarized in Table 7. As can be seen, FlashVSR with TC decoder achieves higher user preference compared to prior approaches, and its performance is comparable to that of the original WAN decoder (Ours-Full).

## D  LLM Usage Statement

We used a large language model(LLM) only to refine grammar and conciseness in some paragraphs. LLM was not involved in formulating research ideas, designing methods, conducting experiments, analyzing results, or drawing conclusions. All scientific contributions, insights, and claims presented in this paper are entirely our own.

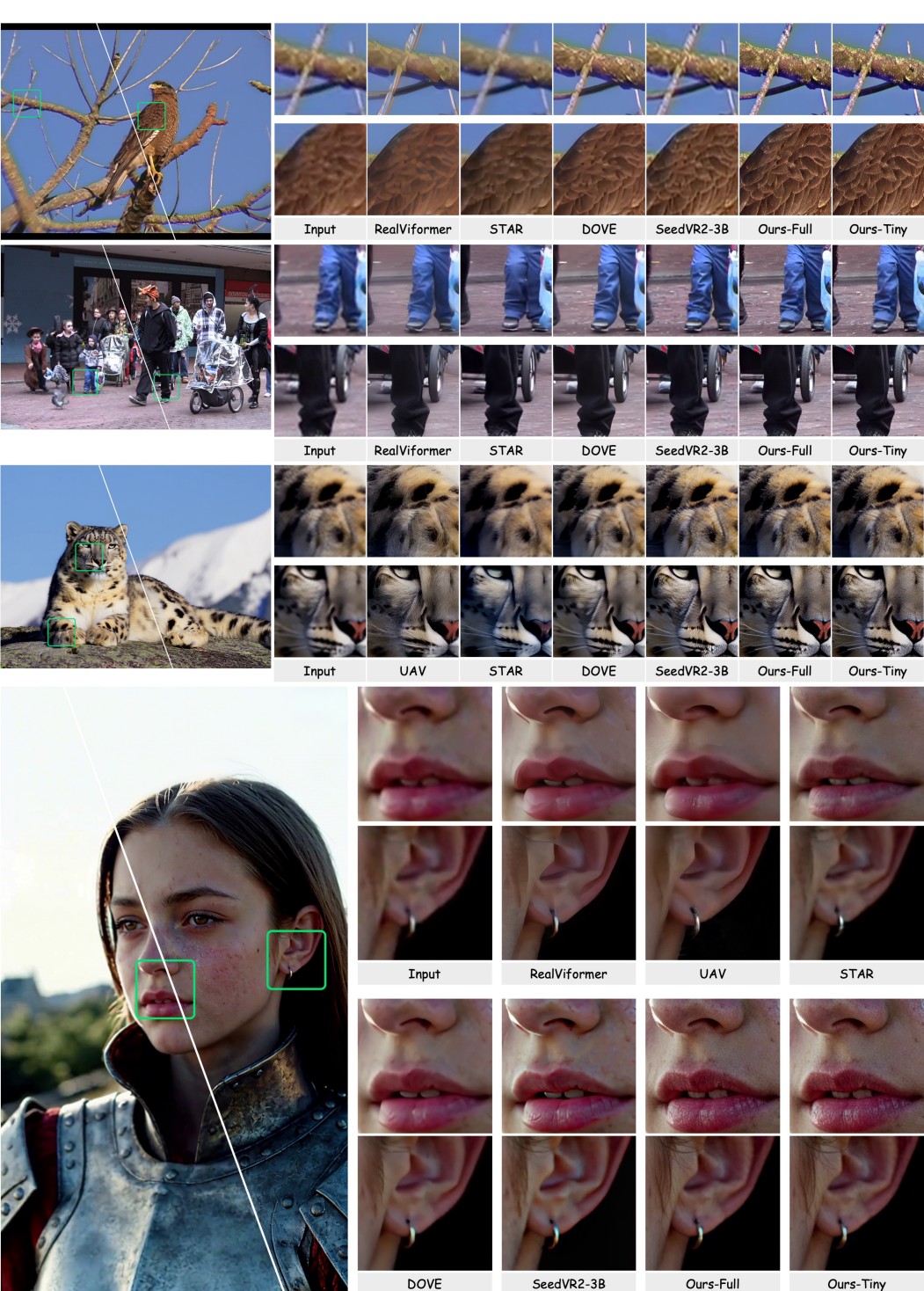

Figure 9: Additional visualization results of video super-resolution on real-world and AIGC videos.

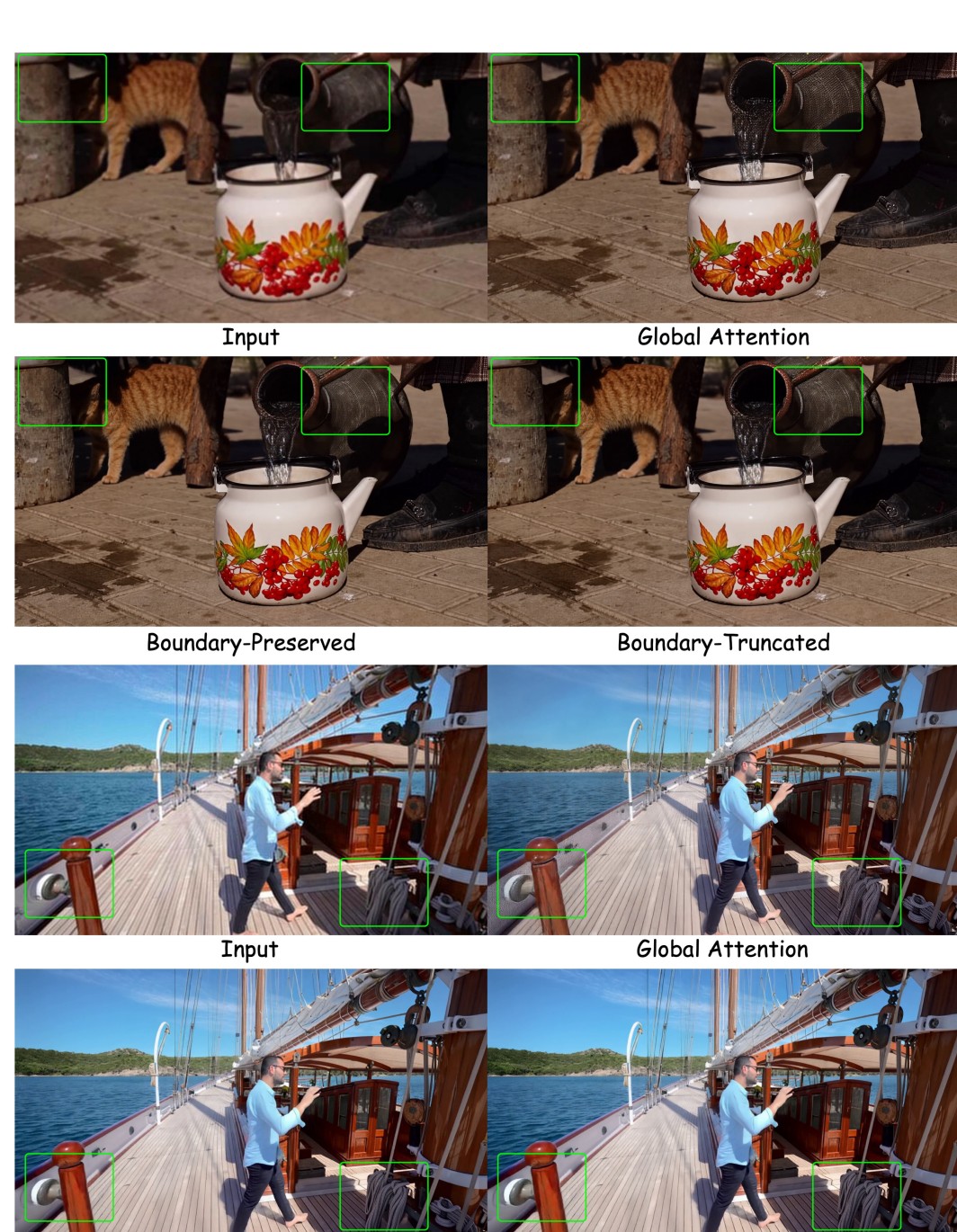

Figure 10: Additional visualization results on high-resolution video super-resolution.

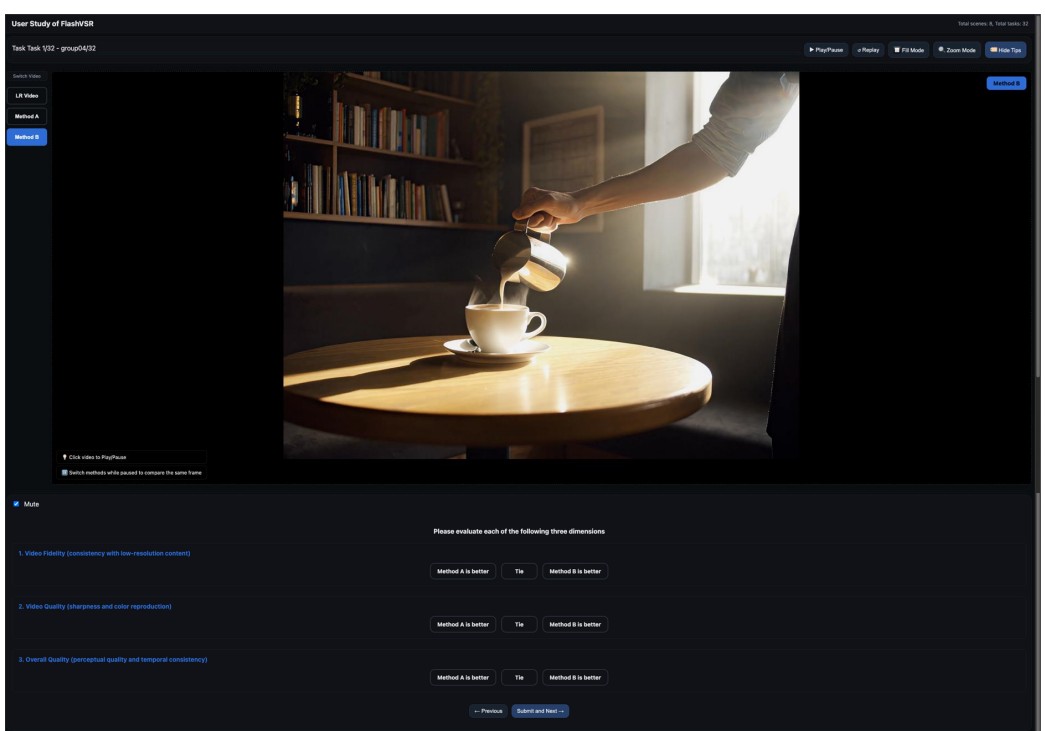

Figure 11: Screenshot of the user study interface for subjective evaluation.

Table 7: User study results (GSB scores, in %) for five one-step VSR models on 32 test sets, including both real-world and AIGC-degraded videos. Higher values indicate stronger user preference.

| Method | Overall Quality ↑ | Video Fidelity ↑ | Video Quality ↑ |
| --- | --- | --- | --- |
| Ours-Tiny | 0.0% | 0.0% | 0.0% |
| Ours-Full | 2.2% | -3.1% | 6.4% |
| SeedVR2-3B | -33.1% | -20.3% | -40.5% |
| DOVE | -30.2% | -14.5% | -34.5% |
| RealViformer | -44.9% | -17.6% | -49.2% |

