# OpenReview forum: "FlashVSR: Towards Real-time Diffusion-Based Streaming Video Super-Resolution"
_ICLR.cc/2026/Conference — ICLR 2026 Conference Withdrawn Submission_

### Official Review · Reviewer_yxVz · 2025-10-23

**Soundness:** 3
**Presentation:** 2
**Contribution:** 3
**Rating:** 6
**Confidence:** 4

**Summary:**

This paper introduces FlashVSR, a diffusion-based framework designed for fast video super-resolution (VSR). It achieves around 17 fps on 768×1408 videos using a single A100 GPU through three key innovations: a three-stage distillation pipeline for streaming SR, locality-constrained sparse attention for efficiency and scalability, and a lightweight conditional decoder for fast reconstruction. The authors also present a new large-scale dataset, VSR-120K, to support training. Experiments demonstrate that FlashVSR achieves state-of-the-art performance with up to 12× speedup over prior diffusion-based VSR methods.

**Strengths:**

1. Proposes a three-stage training pipeline for fast video super-resolution, enabling the method to run at approximately 17 FPS on 768×1408 videos.
2. Adopts a block-sparse causal attention mechanism to reduce the computational cost of bidirectional full attention.
3. Designs a lightweight decoder to further accelerate the overall inference process.
4. Introduces a large-scale VSR dataset to support training and evaluation.

**Weaknesses:**

1. The performance of the full model is inferior to other state-of-the-art methods on key metrics, i.e., PSNR and SSIM.
2. The authors claim that the proposed VSR model supports streaming inference. However, it appears that only the sampling process operates in a streaming manner, while the decoder functions as a sequence-in, sequence-out module. Moreover, latency analysis under the streaming scenario should be discussed in more detail.
3. The supplementary videos mainly showcase VSR results on relatively simple cases such as animals. More comparisons with other SOTA methods on challenging samples, e.g., human videos where identity preservation is difficult, would strengthen the paper.
4. In practice, achieving 17 FPS cannot be strictly considered real-time performance.

Overall, some of the techniques proposed in this paper are similar to, or derived from, previous works such as DMD. Nevertheless, the authors have effectively adapted these techniques to the VSR task and designed a well-structured three-stage training pipeline to achieve a fast and efficient model. I believe this paper makes a valuable contribution to the computer vision community.

**Questions:**

1. Have you tried using a VAE encoder rather than the proposed LR Proj-In to see what the model will perform?

---

### Official Review · Reviewer_Eu1r · 2025-10-29

**Soundness:** 3
**Presentation:** 3
**Contribution:** 2
**Rating:** 4
**Confidence:** 5

**Summary:**

This paper proposes FlashVSR, a diffusion-based video super-resolution (VSR) model optimized for fast and low-memory inference. The main contributions are:

- A three-stage training scheme for one-step VSR model, i.e., first training a joint image-video VSE with full attention. Then, finetuning it into a block-sparse model. Finally convert it into a one-step model with ditillation.

- Using sparse attention with locality constraints for efficiency improvement.

- Further speeding up the efficiency with a tiny decoder.

- Collecting a large-scale VSR dataset for training.

**Strengths:**

- The paper presents a good practice for speeding up VSR via combining existing technologies, with high efficiency compared with existing baselines.

- The paper is easy to follow with clear designs on each component of the model.

- The collected large-scale VSR dataset could facilitate future research towards this direction.

**Weaknesses:**

- The paper lacks theoretical novelty. The proposed method seems to be more like a combination of existing successful technologies, including a multiple-stage training scheme, sparse attention, and DMD loss for distillation.

- While sparse attention is more resource-friendly than full attention. It suffers from limited robustness in terms of the field of image and video generation, reasoning, etc. While the paper claims that the proposed approach shows superior performance with high efficiency, it lacks further verification on this point. After all, there is no free lunch.

**Questions:**

My main concerns are as follows:

1. The weaknesses mentioned above.

2. The paper presents several intuitive claims, lacking detailed analysis and proof to support these claims, e.g., 1) Line 244: ``This reduces attention cost to 10–20% of the dense baseline without performance loss.`` While the metrics may not decrease, the robustness of the model may be weakened.  2) Line 266  ``The core insight is that, unlike video generation, VSR is strongly conditioned on LR frames, so clean historical latents are unnecessary for motion plausibility.`` It is unclear how such insight is obtained, even after I read the discussion in the appendix.

3. While the paper claims that ``The core insight is that, unlike video generation, VSR is strongly conditioned on LR frames, so clean historical latents are unnecessary for motion plausibility.``, it is unclear how to ensure the temporal consistency of the generated results without relying on historical results.

4. The proposed approach relies on the most recent generative prior, i.e., Wan2.1, while most of the compared baselines rely on old generative priors such as CogVideoX, Stable Diffusion upscaler, etc, which are both less efficient and effective. It is unclear how much improvement comes from the proposed scheme rather than the enhancement of the generative prior. Similarly, how the proposed dataset contributes to the final performance is unclear.

5. The distillation scheme used in the paper may lead to the student model’s quality being upper-bounded by the teacher’s.

6. What are the differences between the blue and orange DiT layers in Figure 2?

---

### Official Review · Reviewer_U2gG · 2025-10-31

**Soundness:** 3
**Presentation:** 3
**Contribution:** 2
**Rating:** 4
**Confidence:** 4

**Summary:**

This paper presents FlashVSR, a diffusion-based framework that achieves real-time, streaming video super-resolution (VSR) by addressing the high latency, computational cost, and poor scalability of prior diffusion models. FlashVSR introduces a three-stage distillation pipeline (joint video–image training, sparse-causal adaptation, and one-step distillation), a locality-constrained sparse attention mechanism to bridge the train–test resolution gap, and a tiny conditional decoder that accelerates reconstruction without quality loss. Supported by the newly curated VSR-120K dataset, FlashVSR attains state-of-the-art performance with up to 12× faster inference than previous one-step diffusion methods and maintains strong visual fidelity on ultra-high-resolution videos, demonstrating both practical efficiency and scalability for real-world deployment.

**Strengths:**

1. The paper demonstrates strong originality by being the first diffusion-based one-step framework designed for real-time, streaming video super-resolution (VSR), effectively bridging the gap between diffusion models’ generative quality and practical efficiency.
2. Its three-stage distillation pipeline and locality-constrained sparse attention represent creative and technically sound innovations that address core challenges of latency, scalability, and high-resolution generalization. The proposed tiny conditional decoder further improves efficiency with minimal quality loss, showcasing careful engineering and empirical rigor.
3. The paper’s technical quality is high, with comprehensive experiments, strong baselines, and clear ablations validating each component.

**Weaknesses:**

1. The evaluation scope could be expanded—most experiments focus on perceptual quality and runtime, but there is limited analysis of temporal consistency metrics or failure cases in challenging motion scenarios, which are critical for sparse attention.
2. Although the three-stage distillation and locality-constrained sparse attention are well-motivated, the paper provides limited theoretical justification or ablation on the distillation stages themselves (e.g., contribution of each stage to convergence and fidelity). Additionally, FlashVSR’s reliance on the WAN 2.1 backbone raises questions about generality: it is unclear whether similar results would hold for smaller or alternative diffusion transformers.
3. Though the paper claims real-time performance, this is reported only for A100 GPU; further profiling on different hardware (RTX 3090/4090) would strengthen claims of practical scalability.

**Questions:**

I hold a rating of 4 for now (because there is no 5 option), and I'm glad to raise my score if the weakness and the following questions can be addressed.

1.	Temporal consistency evaluation: The paper primarily reports frame-wise and perceptual metrics (e.g., MUSIQ, CLIPIQA, DOVER). Could the authors provide results using temporal metrics such as Warp Error ?
2.	Scalability and hardware efficiency: The reported 17 FPS performance is based on a single A100 GPU. Could the authors discuss or measure performance on commodity GPUs (e.g., RTX 4090 or mobile chips) to substantiate claims of practical real-time deployment?
3.	Trade-off between sparsity and quality: The paper uses 13.6% attention sparsity. How sensitive is the model’s performance to this sparsity level? Is there a principled way to select it for new resolutions or hardware constraints?

---

### Official Review · Reviewer_XJux · 2025-10-31

**Soundness:** 2
**Presentation:** 3
**Contribution:** 2
**Rating:** 4
**Confidence:** 3

**Summary:**

The paper introduces a novel diffusion-based framework for Video Super-Resolution (VSR). The proposed model handlesthe high latency and computational cost associated with diffusion models. The authors incorporate sparse attention mechanisms and a one-step conditional decoder to improve efficiency. Experimental results demonstrate that the proposed method outperforms existing approaches across multiple datasets and evaluation metrics.

**Strengths:**

* The authors collected a new dataset, VSR-120, specifically designed for the video super-resolution task.
* Both qualitative and quantitative evaluations indicate that the proposed approach achieves superior performance compared to prior work.
* The incorporation of block-sparse attention effectively reduces computational complexity while preserving reconstruction quality.

**Weaknesses:**

* The paper does not explain the contribution of the new dataset to the model’s performance. It remains uncertain how much the reported improvements depend on the characteristics or quality of VSR-120.
* In Table 1, the results reported for Upscale-a-Video are worse than those in the original publication. For example, in the original paper, YouHQ achieves PSNR 25.83, SSIM 0.73, and LPIPS 0.269, and SPMCS reports PSNR 25.32, SSIM 0.741, and LPIPS 0.222. Similar discrepancies appear for VideoLQ and AIGC30. The current version of the paper does not explain these inconsistencies between the reproduced results and those originally reported in Upscale-a-Video.
* Several architectural and implementation details are insufficiently described, which makes it difficult to assess the proposed method (see Questions below) fully.

**Questions:**

* How is the low-resolution (LR) Proj-In layer implemented in practice?
* What is the motivation for including the LPIPS metric as part of the decoder’s reconstruction objective?
* What is the rationale for combining DDM with flow matching, and how does this integration enhance model performance?
* Where does the conditional tiny encoder fit into the FlashVR architecture after training?
* What is the impact of varying the inference resolution range, and how was the local attention window configured during inference?
* In Figure 2, what do the “Real” and “Fake” scores represent?

---

### Note · Authors · 2025-11-13

**Comment:**

I have read and agree with the venue's withdrawal policy on behalf of myself and my co-authors.

**Withdrawal Confirmation:**

I have read and agree with the venue's withdrawal policy on behalf of myself and my co-authors.